# Role of Kinins in Hypertension and Heart Failure

**DOI:** 10.3390/ph13110347

**Published:** 2020-10-28

**Authors:** Suhail Hamid, Imane A. Rhaleb, Kamal M. Kassem, Nour-Eddine Rhaleb

**Affiliations:** 1Hypertension and Vascular Research Division, Department of Internal Medicine, Henry Ford Hospital, Detroit, MI 48202, USA; Hsuhail1@hfhs.org (S.H.); irhaleb1@hfhs.org (I.A.R.); 2Division of Cardiology, Department of Internal Medicine, University of Louisville Medical Center, Louisville, KY 40202, USA; kamkassem@gmail.com; 3Department of Physiology, Wayne State University, Detroit, MI 48201, USA

**Keywords:** kallikrein–kinin system, bradykinin, B_1_ and B_2_ receptors, angiotensin-converting enzyme, angiotensin receptor blockers, hypertension, myocardial infarction, mice

## Abstract

The kallikrein–kinin system (KKS) is proposed to act as a counter regulatory system against the vasopressor hormonal systems such as the renin-angiotensin system (RAS), aldosterone, and catecholamines. Evidence exists that supports the idea that the KKS is not only critical to blood pressure but may also oppose target organ damage. Kinins are generated from kininogens by tissue and plasma kallikreins. The putative role of kinins in the pathogenesis of hypertension is discussed based on human mutation cases on the KKS or rats with spontaneous mutation in the kininogen gene sequence and mouse models in which the gene expressing only one of the components of the KKS has been deleted or over-expressed. Some of the effects of kinins are mediated via activation of the B_2_ and/or B_1_ receptor and downstream signaling such as eicosanoids, nitric oxide (NO), endothelium-derived hyperpolarizing factor (EDHF) and/or tissue plasminogen activator (T-PA). The role of kinins in blood pressure regulation at normal or under hypertension conditions remains debatable due to contradictory reports from various laboratories. Nevertheless, published reports are consistent on the protective and mediating roles of kinins against ischemia and cardiac preconditioning; reports also demonstrate the roles of kinins in the cardiovascular protective effects of the angiotensin-converting enzyme (ACE) and angiotensin type 1 receptor blockers (ARBs).

## 1. Kallikrein–Kinin System

Plasma and tissue kallikreins are potent enzymes that generate kinins by hydrolyzing kininogens, which circulate at high concentrations in plasma (Figure 1). Kinins are rapidly destroyed by kininases [1].

Plasma kallikrein, also known as Fletcher factor, is expressed mainly in the liver; in plasma it is found in the zymogen form (pre-kallikrein) and differs from glandular kallikrein not only biochemically but also immunologically and functionally. Plasma kallikrein is encoded by a single gene, KLKB1. Some polymorphisms of this gene are associated with end-stage renal disease and hypertension [4,5,6]. Plasma kallikrein preferentially releases bradykinin (BK) from high-molecular-weight kininogen (HMWK), also known as the Fitzgerald factor. Together with HMWK and Hageman factor (factor XII), plasma kallikrein is involved in coagulation and fibrinolysis. The plasma kallikrein–HMWK system, acting through the release of BK, could be involved in the local regulation of blood flow and in mediating some of the effects of angiotensin-converting enzyme (ACE) inhibitors. On the other hand, patients with a congenital deficiency of plasma HMWK (Fitzgerald trait) have normal amounts of kinins in their blood [7] (For a review of the plasma kallikrein–HMWK system, see [8,9,10,11]).

### 1.1. Tissue (Glandular) Kallikrein

Kallikreins (KLK) are serine proteases with very high homology and are expressed by genes that are compactly clustered and arranged in tandem on the same chromosome. The kallikrein family is estimated to contain at least 15 genes in humans, 20 in rats and 23–30 in mice [12]. However, not all these proteases generate kinins, despite their highly homologous amino acid composition; rather, they act on different substrates and are expressed in different tissues [13,14,15,16]. KLK1 (tissue kallikrein) is encoded by a single gene containing five exons and four introns. While the KLK1 gene is expressed in the submandibular gland, pancreas and kidney, small amounts of kallikrein mRNA were detected in the heart, vascular tissue and adrenal glands (PCR) [17,18]. Kallikrein and similar enzymes have been found in the arteries and veins [19], heart [20], brain [21], spleen [22], adrenal glands [23] and blood cells [18]; they have also been observed in the pituitary gland [24,25], pancreas [26], large and small intestines [27,28], and salivary and sweat glands [29] along with their exocrine secretions. Tissue kallikrein immunoreactivity can be found in plasma, primarily in the inactive form; only a small portion remains active [30,31,32,33]. Approximately 50% of urinary kallikrein is found to be inactive (zymogen) in humans [34] and rabbits [35], while in rats most of it is active [36]. Tissue kallikrein can release kinins from low-molecular-weight kininogen (LMWK) and HMWK. In humans, KLK1 releases lys-bradykinin (kallidin; KD), whereas in rodents it releases bradykinin [37,38].

### 1.2. Kininogens or Kallikrein Substrates

Kininogens or kallikrein substrates are the precursors of kinins. In plasma there are two main forms, LMWK and HMWK. Interestingly, the human genome contains a single copy of the kininogen family, whereas three copies exist in the rat (one encoding K-kininogen and two encoding T-kininogen, and two homologous kininogen genes in the mouse genome, mkng1 and mkng2 [39,40]. In rats, T-kininogen, the major kininogen, releases T-kinins (Ile-Ser-bradykinin) when incubated with trypsin but not with tissue or plasma kallikrein. T-kinin, acting via B_2_ receptors, is one of the main acute reactants of inflammation [41,42,43,44]. HMWK is involved in the early stages of surface-activated coagulation (intrinsic coagulation pathway) [8,10,45].

### 1.3. Kinins

Kinins are oligopeptides that contain the sequence of bradykinin and act mainly as local hormones, since they circulate at very low concentrations (1 to 50 fmol/mL) and are rapidly hydrolyzed by kininases. However, they exist in higher concentrations in the kidney, heart and aorta (100 to 350 fmol/g), further supporting the hypothesis that in these tissues, they act mainly as local hormones [46]. Eicosanoids, nitric oxide (NO), endothelium-derived hyperpolarizing factor (EDHF), tissue plasminogen activator (T-PA) and cytokines reportedly mediate some of the effects of kinins [47,48,49,50] (Figure 2).

### 1.4. Kininases

Kininases are peptidases found in blood and other tissues that hydrolyze kinins and other peptides [51]. The most well-known is the angiotensin-converting enzyme (ACE) or kininase II, which converts angiotensin I to II and inactivates kinin, N-acetyl-seryl-aspartyl-lysyl-proline (Ac-SDKP), substance P and other peptides [51,52]. Another important kininase is neutral endopeptidase 24.11 (NEP-24.11), also known as enkephalinase or neprilysin, which not only hydrolyzes kinins and enkephalins but also destroys atrial natriuretic peptide (ANP), brain natriuretic peptide (BNP) and endothelin [53,54]. Our research suggests that it may be an important renal kininase, at least in rats [55]. When ACE, NEP-24.15, aminopeptidases and carboxypeptidases are suppressed in vivo, endogenous plasma kinins do not increase significantly and their half-life remains less than 20 s, suggesting that other peptidases are also important for kinin metabolism [56]. In addition, several other kininases have been described, including carboxypeptidase N (CPN) and carboxypeptidase M (CPM), together called kininase I [11,57,58]. These enzymes are membrane-bound proteins that cleave C-terminal Arg or Lys residues from peptides and proteins, and are responsible for the conversion of BK or KD into B_1_ receptor ligands des-Arg^9^-BK or des-Arg^10^-KD, respectively [59,60,61]. Recently, a study has shown that Kinin B_1_R positively modulates both CPM expression and activity, suggesting that CPM–B_1_R interaction in membrane micro-domains might affect enzyme activity, beyond interfering in receptors signaling [62]. However, the physiological meaning of such interactions remains to be elucidated, especially that CPM could cleave C-terminal Arg or Lys from many other peptides or proteins such as the release of fribrinopeptide B_15–42_ [61].

### 1.5. Receptors

Kinins act on two well defined and characterized receptors, namely B_1_ and B_2_ [63,64]. Both have been cloned and belong to the family of 7-transmembrane receptors linked to G-proteins [65]. B_1_ receptors are present at very low density (or not at all) in normal tissue but are expressed and synthesized de novo during tissue injury, inflammation and administration of lipopolysaccharides such as endotoxin [59,60]. Their main agonists are des-Arg^9^-bradykinin and des-Arg^10^-kallidin. B_2_ receptors, the main receptors for BK and KD, mediate most of the effects of BK [58,66].

In humans, the B_2_ receptor is reportedly activated directly by kallikreins and other serine proteases since this effect can be blocked by the potent and specific B_2_ receptor antagonist, icatibant [67]. Moreover, B_2_ has been found to interact directly with AT_2_ [68], B_1_ [69], ACE [70], and even other B_2_ receptors (forming homodimers) [71]. However, the physiological and pathophysiological significance of such receptor interactions remain unknown.

## 2. The KKS in the Vasculature and Regulation of Local Blood Flow

Arteries and veins contain a kallikrein-like enzyme, and both vascular tissue and smooth muscle cells in culture are known to express kallikrein mRNA [17,19]. Vascular smooth muscle cells in culture release both kallikrein and kininogen [72]. Thus, the components of the KKS are present in vascular tissue, where they could play an important role in regulation of vascular resistance. Arteries isolated from mice lacking the kallikrein gene reportedly exhibited significantly reduced flow-induced dilatation compared to controls, suggesting that the KKS in the arterial wall participates in the regulation of local blood flow [73,74]. Moreover, in humans a partial genetic deficiency of tissue kallikrein (R53H) was associated with inward remodeling of the brachial artery that renders it incapable of adapting to a chronic increase in wall shear stress, a form of arterial dysfunction that affects 5–7% of Caucasians [75]. In organs rich in kallikrein, such as the submandibular gland, uteroplacental complex and kidney kinins play an important role in local regulation of blood flow [76,77,78,79]. In nephrectomized pregnant rabbits infused with an angiotensin receptor antagonist, ACE inhibitors increased both uterine and placental blood flow and also raised levels of immunoreactive PGE_2_; subsequently, all of these effects were blocked by a kinin antibody [77].

## 3. Kinins in Regulation of Cardiac and Renal Blood Flow

Kinins play an important role in regulating renal blood flow. In sodium-depleted dogs, the infusion of low-dosed kinin antagonist into renal artery blocked renal kinins, which decreased renal blood flow and autoregulation of the glomerular filtration rate (GFR) without altering blood pressure [80]. The role of kinins in the regulation of renal blood flow distribution was determined using a laser-Doppler flowmeter [78]. Hence, papillary blood flow, but not outer cortical blood flow, could be reduced by a kinin receptor antagonist, suggesting that intrarenal kinins are important for the inner medulla blood flow regulation. In anesthetized rats, blocking kinins decreases renal blood flow [79]. In dogs, when kallikrein excretion was stimulated by sodium deprivation, intrarenal administration of BK receptor antagonist partially blocked the effect of enalaprilat on renal blood flow [81], suggesting that both the blockade of the renin–angiotensin system (RAS) and the increase in endogenous kinins accounted for the increased renal blood flow caused by ACE inhibition. In normal rats, kinins play a minor role in regulation of renal blood flow; however, when the KKS is stimulated by low sodium intake or mineralocorticoids, or when endogenous kinin degradation is inhibited, kinins participate in the regulation of renal blood flow [82,83,84].

Numerous experimental studies using animal models have focused the role of the KKS in the regulation of coronary blood flow and its repercussion on cardiac diseases. Local cardiac KKS is believed to exert a significant cardiac protective role by delaying the development of heart failure such as in myocardial infarction. This has been demonstrated by using strategies such as kallikrein gene transfer, tissue kallikrein infusion, and human kallikrein over expressing animals, or B_1_ or B_2_ BK receptor knockout mice. Studies have suggested that the KKS increases coronary blood flow, and decreases infarct size and left ventricular remodeling post myocardial infarction [85,86]. Some of these aforementioned studies have also shown that the beneficial effects seen in the treatment with ACE inhibitors and/or angiotensin receptor blockers (ARBs) are not only due to the inhibition of Ang II generation or effects, but also in part to the prevention of bradykinin enzymatic degradation; this suggests that the KKS plays a significant role in the effects of ACE inhibition, particularly on angiogenesis and myocardial regeneration [87,88,89,90,91].

## 4. Kinins in Blood Pressure Regulation and the Pathogenesis of Hypertension

Proper blood pressure regulation and function maintains the balance between vasopressor and vasodepressor systems. Any alterations of this equilibrium may result in (a) hypertension, (b) target organ damage, (c) ineffective antihypertensive treatment, or (d) hypotension and shock. These alterations could occur because of (a) modification in the genetic factors such as a mutation in one or more genes of the vasoactive system, (b) environmental factors that alter the activity of vasoactive systems and/or epigenetic factors. The role of the KKS in the pathogenesis of hypertension has been studied by (1) measuring various components of the system, (2) examining bradykinin B_2_ receptor antagonists, (3) studying mice with B_1_, B_2_, or both deleted by homologous recombination, (4) the deletion of the tissue kallikrein gene, and (5) observing rats deficient in kininogen. Endocrine and neuroendocrine vasopressor systems, such as the RAS system and catecholamines, have been long established as important endogenous regulators of blood pressure, the pathogenesis of some forms of hypertension, and end organ damage and dysfunction. On the other hand, the role of vasodepressor systems remains controversial; however, there is some evidence that suggests vasodepressor systems may be important contributors in regulation of blood flow, renal function, the pathogenesis of salt-induced hypertension, and end organ injury and the cardio-renal protective effects of ACE inhibitors and ARBs [89,92,93,94,95]. Vasodepressor hormones such as kinins, eicosanoids, NO, carbon monoxide (CO), and EDHF are some of the local hormones that may oppose the effects of vasopressor systems. Some vasodepressors such as atrial (ANP), brain (BNP), and C-type (CNP) natriuretic peptides may act as both endocrine and local hormones.

Decreased activity of the KKS may play a role in hypertension. Indeed, low urinary kallikrein excretion in children is one of the major genetic markers associated with a family history of essential hypertension, and children with high urinary kallikrein are less likely to be genetically predisposed to hypertension [96,97,98,99]. A restriction fragment length polymorphism (RFLP) for the kallikrein gene family in spontaneously hypertensive rats (SHRs) has been linked to high blood pressure [100], and urinary kallikrein excretion is decreased in several models of genetic hypertension and in renovascular hypertension [101,102,103,104]. Decreased urinary kallikrein in (a) normotensive children of patients with essential hypertension, (b) genetically hypertensive rats and (c) pre-hypertensive Dahl salt-sensitive rats [105,106,107,108,109] could be secondary to hypertension through mechanisms that might be specific for each model.

Blood pressure and cardiovascular function are normal in HMWK-deficient rats, and B_1_^-/-^ or B_2_^-/-^ and tissue kallikrein^-/-^ mice, however, in the kallikrein^-/-^ mice, the structure and function of the heart are clearly abnormal [74,110,111,112,113]. Chronic blockade of B_2_ receptors with the icatibant (Hoe-140, B_2_ antagonist) did not increase blood pressure under normal conditions or in situations that favor hypertension in rats [111,114]. However, contradicting findings have been published, reporting that lack of circulating kininogen or blockade of B_2_ receptors are associated with significant increases in blood pressure under normal conditions or when animals are challenged with a pressor agent such as a high salt diet or Ang II infusion [115,116,117,118].

The bradykinin B_2_^-/-^ mice have normal blood pressure; however, they develop hypertension when fed a very high-sodium diet (8%) for at least 2 months [119,120]. Thus, low kinin activity may be involved in the development and maintenance of salt-sensitive hypertension. However, in B_2_^-/-^, mice hypertension was not exacerbated when induced by mineralocorticoids (renin-independent) or coarctation of the aorta (renin-dependent) [113]. Additionally, others have reported that as these mice grow older, they also develop hypertension and left ventricular hypertrophy even on a normal sodium diet [121,122,123]. Others have shown that mice lacking the gene for B_2_ receptors (B_2_R(-/-)*^CRD^* mice) exhibited transient hypertension phenotype from 2 to 4 months of age, but developed salt diet-dependent hypertension [124]. However, we and others were unable to confirm that B_2_ ablation renders mice spontaneously hypertensive [110,113,120,125,126]. Mice deficient in B_1_, B_2_ or both, as well as mice with low tissue kallikrein, had blood pressure readings similar to wild-type controls, confirming that kinins are not essential for the regulation of basal blood pressure [126].

A lack in both B_1_ and B_2_ (as in Akita mice) exacerbates diabetic complications as well as oxidative stress, mitochondrial DNA damage and overexpression of fibrogenic genes, yet, these mice are normotensive [127]. In kininogen-deficient Brown Norway Katholiek rats (BNK), administration of mineralocorticoids and salt or angiotensin II increased blood pressure to the same degree as rats with a normal KKS [111], contradicting reports by other investigators [115,116,117]. Thus, taken together, the published data would suggest that kinins are not critical for blood pressure regulation, nor are they required for the development of hypertension, except for animals under a very high salt diet. Thus, a chronic blockade of the KKS does not cause hypertension. There are in the literature some fine reviews depicting the role of kinins in hypertension and cardiovascular regulation (please refer to [11,58,128,129].

KKS could also have an impact on blood flow and pressure via bradykinin, which has been demonstrated to enhance transmitter release from the sympathetic nerves. Indeed, it was first discovered that bradykinin potentiates the release of adrenaline from the adrenal medulla [130]. Moreover, bradykinin was found to potentiate the release of norepinephrine from mouse, rat, and human right atria; however, the opposite is true for rabbit heart in which bradykinin inhibits norepinephrine release [131,132,133]. In addition, Kansui et al. reported that bradykinin enhances the sympathetic purinergic neurotransmission via presynaptic B_2_ receptors in rat mesenteric resistance arteries [134]. However, the physiological and clinical significance of the bradykinin on the sympathetic nervous system remain unclear and warrant further investigation.

## 5. Role of Kinins in Thermoregulation

Various contributors and mechanisms participate in the maintenance of thermoregulatory homeostasis in individuals that are exposed to environmental temperatures. The primary physiological responses include an increase in metabolism (shivering thermogenesis), an alteration in the vasomotor responses (peripheral vasoconstriction/vasodilation), and a circulatory response (countercurrent heat mechanism). These factors added to fitness level, body composition, age, gender, and ethnicity could influence an individual’s ability to regulate body temperature [135]. Particularly, it has been established that Caucasians markedly exhibit a greater expansion of energy to maintain their core temperature in response to acute cold stress as compared to African-American subjects. Caucasian individuals are also at reduced risk for the development of hypothermia compared to African-American subjects, as demonstrated by the increased shivering thermogenesis and energy expenditure, which helps maintain temperature homeostasis [135]. Kallikrein, the enzyme responsible for the release of kinins, is diminished in African-Americans as demonstrated by the significant decrease in renal kallikrein and potassium excretion [136]; also, Allelic frequencies for three of the four polymorphisms of the B_2_ receptor gene were significantly different from those reported in Caucasian populations. Among the polymorphisms analyzed, a potentially and functionally significant polymorphism in the core promoter of the kinin B_2_ receptor (C-58-->T transition) [137] has been observed. Thus, this B_2_ receptor promoter polymorphism may represent a susceptibility marker for not only essential hypertension in African Americans, but also their lack of efficient thermoregulation. Mice in which the gene expressing B_2_ receptor has been specifically deleted from the endothelium (B_2_^flox/flox^.Tie2^Cre^) presented normal blood pressure readings compared to the wild type. However, B_2_^flox/flox^.Tie2^Cre^ mice experienced lower body temperature (by about 1.5 °C) compared to wild-type mice when housed in a room at 23 °C, which is 7 °C below thermoregulation (N.-E. Rhaleb, unpublished observation). On the other hand, B_1_ receptors, which are induced in inflammatory diseases such as type I diabetes, could also contribute to hyperthermia through a vagal sensory mechanism involving prostaglandins (via Cyclo-oxygenase-2) and nitric oxide [138]. Nevertheless, more studies are ongoing to determine this novel role of endothelial B_2_ receptors under basal and stress conditions such as cold and hot environments and in the hypertensive state.

## 6. KKS Versus SARS-CoV2 in COVID-19 Patients

The COVID-19 pandemic has taken the world by storm and has quickly become a major morbidity risk factor and a source of mortality. Numerous clinical observations indicate that COVID-19 fatalities were linked not only to respiratory distress but multifaceted cardiac involvement including myocarditis, hypoxia induced type 2 myocardial infarction, acute atherothrombotic myocardial infarction, cardiac injury from drug toxicity, and endogenous catecholamine adrenergic activity that could lead to the development of stress cardiomyopathy and cardiac arrhythmias. Millions of patients have tested positive for the SARS-CoV-2 virus, the virus responsible for the COVID-19 disease, with up to a 3.7% death toll, a rate that continues to increase as more individuals are tested. Clinical data have indicated that 20–36% of patients with COVID-19 are afflicted by acute myocardial injury [139], and this incidence rate will certainly be changing as and when new epidemiological and clinical data are published on the effects of SARS-CoV2 virus on the CVD of patients with or without existing comorbid factor. New onset of heart failure (HF) was observed in as much as a quarter of hospitalized COVID-19 patients; and in as much as one-third of those admitted to the intensive care unit [140]. Therefore, it is thought that there must be a host response to the Severe Acute Respiratory Syndrome Coronavirus 2 of the genus Betacoronavirus (SARS-CoV2) during which the innate pro-inflammatory immune response is triggered. Angiotensin-converting enzyme (ACE)-2 surfaced as an important receptor for the virus, which permits viral cell entry and propagation [141,142,143,144]. During this COVID-19 crisis, scientists have discovered that SARS-CoV-2 uses ACE2 as a receptor for entry in the host cells, and the serine protease TMPRSS2 for S protein priming [145,146,147,148]. Figure 3 summarizes the relationships between ACE2 and RAS. In addition to the involvement of ACE2 in the conversion of Ang I into Ang (1–9) or Ang II into Ang (1–7) [149,150], ACE2 inactivates des-Arg^9^-BK (the B_1_ receptor agonist) [151,152], and thus, provides anti-inflammation effects. The B_1_ agonist is responsible for the potent and sustained pro-inflammatory and hyperalgesia effects via B_1_ receptors [11,129]. Moreover, it has been hypothesized that the virus-mediated down-regulation of ACE2 causes a burst of inflammatory cytokine release through dysregulation of the RAS (ACE/Ang II /AT_1_R axis), attenuation of ACE2/MasR axis, increased activation of desArg^9^-BK/B_1_ receptor pathway, and activation of the complement system including C5a and C5b-9 components [152]. Moreover, Ang (1–7) acting through Mas receptors or Ang (1–9) through AT_2_ receptors activates tissue KKS to release kinins, thus providing cardiovascular and renal protection that are mediated by B_2_ receptors [2,153,154,155]. However, a recent clinical study has interestingly reported that COVID-19 patients that received icatibant, a potent B_2_ receptor antagonist, experienced improved oxygenation [156]; this is consistent with the role of B_2_ receptors in mediating the swelling of soft tissues as a result of excess fluid accumulation [156], and making COVID-19 patients face a life-threatening condition whereby the lungs cannot provide the body’s vital organs with enough oxygen [2,11,58,129]. It has been proposed that pulmonary edema could be due to a local vascular problem due to the activation of B_1_ and B_2_ receptors on endothelial cells in the lungs; as a result, the blockade of kinin receptors and/or inhibiting plasma kallikrein activity, could have an ameliorating effect on early disease caused by COVID-19 and might prevent ARDS [156]. However, one must be reminded that icatibant could act as an antagonist of the B_1_ receptor as well, because we have shown that icatibant could be converted to desArg^9^-icatibant and block the effect of desArg^9^-BK [157]. Thus, using experimental models with genetically modified B_1_ or B_2_ receptors could explain the contribution of each of the kinin receptors during SARS-CoV2 exposure. Noteworthy, the serine protease TMPRSS2 for S protein priming has also surfaced as an important protein that facilitates the propagation of SARS-CoV2 virus [145,146,147,148]. Indeed, a TMPRSS2 inhibitor approved for clinical use has been shown to block entry, and thus, together with an ACE2 inhibitor, these inhibitors could constitute a potential treatment option. However, how and whether ACE2 and TMPRSS2 interact, or the balance between ACE2, Angiotensin peptides and kinins during the viral attack remain to be elucidated.

## 7. Role of Kinins in the Therapeutic Effect of ACE Inhibitors and Angiotensin Receptor Blockers (ARBs)

Inhibition of the degradation of kinin and other vasodilator oligopeptides may contribute to the antihypertensive effect of ACE inhibitors. While a blockade of angiotensin II formation plays an important role in this process, the role of kinins or other endogenous peptides such as Ac-SDKP is less well established (Figure 4). Concentrations of kinins in tissue may well exceed blood levels and could conceivably contribute to the anti-hypertensive and vasodilator effects of ACE inhibitors in humans [86,89]. Orally active ACE inhibitors are effective antihypertensive agents, not only in high-renin hypertension but also in clinical and experimental models that do not involve the systemic RAS [158,159]. Thus, some of their effects may be mediated by a local RAS, kinins or some other undetermined mechanism, since ACE can hydrolyze numerous other peptides (Figure 4). ACE inhibitors may also augment the effect of kinins by interacting directly with the B_2_ receptor [160]. Blood kinins are unchanged or moderately increased after treatment with ACE inhibitors [3,161,162] (for a review, see [163,164].) Kinins in the urine reportedly increase more consistently following ACE inhibition therapy, which suggests their renal concentration increases too [55,165,166,167,168], thus strengthening the antihypertensive effect of ACE inhibitors by altering renovascular resistance and increasing sodium and water excretion. Studies involving various experimental models of hypertension have shown that the acute antihypertensive effect of ACE inhibitors is attenuated by blocking kinins with either high titer kinin antibodies [169,170,171] or a B_2_ receptor antagonist [161,162,172].

Kinin antagonists also partially reversed their antihypertensive action in rats with renovascular hypertension [173]; however, lack of B_2_ receptors did not abolish the anti-hypertensive effect of ACE inhibition in mice with renovascular (2 kidney-1 clip or 2K1C) hypertension (Figure 5). This is not surprising, since it is well established that the RAS plays a major role in the development of renovascular hypertension. However, kinins may be responsible for the acute antihypertensive effect of ACE inhibitors such as enalaprilat [162]. Indeed, in rats with severe hypertension induced by aortic ligation between the renal arteries, renin is necessary for the pathogenesis of hypertension [158]; however, acute and severe hypertension can damage the endothelium enough to activate plasma pre-kallikrein and increase kinin formation. Enalaprilat lowered the mean blood pressure by 48 ± 6 mm Hg in the controls and 21 ± 4 mm Hg in the kinin antagonist group (*p* < 0.01); however, kinins in arterial plasma were not significantly altered by the ACE inhibitor (41 ± 10 vs. 68 ± 20 pg/mL). We have also confirmed the role of B_2_ receptor in the acute hypotensive effect of ACE inhibition in a model of glucocorticoid-salt-induced hypertension using B_2_ receptor knockout mice [113]. As indicated earlier, kinins’ concentration in the blood must reach at least 1000 pg/mL before they can efficiently lower blood pressure in non-anesthetized rats [174]. Thus, the effect of the ACE inhibitor may have been due to an increase in tissue kinins, which could regulate vascular resistance acting as a paracrine hormonal system. Cachofeiro et al. [161] demonstrated that pretreatment with a B_2_ receptor antagonist or NO synthesis inhibitor attenuated the acute antihypertensive effect of both captopril and ramipril in SHR whereas a prostaglandin synthesis inhibitor made no difference, suggesting that this effect was due to bradykinin stimulating the release of NO. However, in dogs, kinins may strengthen the acute hypotensive effect of ACE inhibitors via prostaglandins [175].

In humans, an ACE insertion/deletion polymorphism at intron 16 of the ACE gene could be important for bradykinin metabolism [176], as ACE activity is higher in subjects with ACE deletion and correlates with rapid bradykinin degradation. In normotensive subjects and hypertensive patients with low or normal renin, aprotinin (an inhibitor of kallikrein and other proteases) partially blocked the acute antihypertensive effect of captopril [177]. While that could have been due to kinin inhibition, other investigators tested a specific B_2_ kinin receptor antagonist (icatibant) and found that the short-term blood pressure effects of ACE inhibitors were attenuated in both normotensive and hypertensive subjects [178], suggesting that the acute effect of ACE inhibitors is mediated in part by kinins affecting local and peripheral vascular resistance either directly or through release of prostaglandins and NO.

The contribution of kinins to the chronic antihypertensive effects of ACE inhibitors remains controversial. In renovascular hypertension (2K1C), chronic blockade of kinin receptors interferes with ramipril’s ability to lower blood pressure [179]. In mineralocorticoid hypertension, where KKS and ACE activity are reportedly increased [180], chronic ACE inhibitors have a small but significant antihypertensive effect that can be blunted by blocking the B_2_ receptor with icatibant [111,181], suggesting that kinins may be involved; however, they are ineffective in SHR [179] or hypertension that is induced by aortic coarctation [114,161,182]. Therefore, the role of kinins in the long-term antihypertensive effect of ACE inhibitors depends on the model. To our knowledge, no studies of chronic KKS blockade have been conducted in humans.

ACE inhibitors, but not ARBs, are also known to increase N-acetyl-seryl-aspartyl-lysyl-proline (Ac-SDKP), which also promotes anti-fibrosis and anti-inflammation. Ac-SDKP is an endogenous tetrapeptide found in circulation and in various organs, including the heart [183,184]. Ac-SDKP is endogenously produced from a 43-amino acid thymosin 4 (Tβ4) through two successive enzymes, meprin α and prolyl oligopeptidase [185,186,187,188]. On the other hand, Ac-SDKP is hydrolyzed mainly by ACE and its circulating levels were found to increase more than five-fold in patients treated with ACE-I [189]. Studies from our group have shown that in models of hypertension and myocardial infarction (MI), Ac-SDKP exerts anti-inflammatory and anti-fibrotic effects in the heart [190,191,192]. However, whether Ac-SDKP functions could provide additive cardiovascular protective effects to those mediated by first choice pharmacotherapy for cardiac diseases, such as ARBs, ACE-I, β-adrenergic blockers or calcium channel blockers, remains to be elucidated.

## 8. Role of Kinins in the Effects of ACE Inhibitors on Hypertensive Target Organ Damage and in Heart Failure Post-MI

ACE inhibitors have been shown to reverse LV hypertrophy in essential hypertension and in various experimental models of hypertension, in great part due to reduced afterload. Although Linz et al. reported they were able to reverse the anti-hypertrophic effects of an ACE inhibitor using a kinin antagonist [193], we have not been able to confirm this [114]. Nevertheless, there is a large body of evidence that ACE inhibitors reduce morbidity and mortality, improve cardiac function, regress LV remodeling, and prolong life in patients with heart failure (HF), not only improving cardiac function and increasing survival but also lessening myocardial re-infarction [194]. Since ACE inhibitors prevent kinin degradation in the coronary and renal circulation, it could be through advanced pathways that kinins stimulate NO and PGI_2_ (important inhibitors of platelet aggregation) to block platelet aggregation, coronary arterial stenosis and eventually myocardial infarction or renal ischemia. Kinins are also potent stimulators of T-PA [50,195], thereby activating plasmin and fibrinolysis. In a rat model of HF due to MI, ACE inhibitors improved cardiac function and attenuated remodeling, and these beneficial cardiac effects were diminished by blocking kinins [91]. Moreover, in B_2_^-/-^ mice and kininogen-deficient rat post-MI, ACE inhibitors had diminished protective effects [90,95]. Although, despite not exactly knowing how kinins protect the heart, it is possible kinin-stimulated release of NO, EDHF, and/or PGI_2_ could be largely responsible [196,197]. Table 1 summarizes some of the important findings on the putative role of kinins in myocardial infarction. The bradykinin-induced EDHF could be highly relevant in conditions of tilted NO and PGI_2_ of any vascular bed. This results in the maintenance of intact endothelial function in disease states such as hypertension, heart failure and diabetes in which NO-mediated responses are compromised due to increased oxidative stress [198,199]. EDHF has also been shown to mediate bradykinin-induced mouse ductus arteriosus patency when NO, PGI_2_ and carbon monoxide have been suppressed [200]. These combinatorial factors contribute substantially to basal human forearm vascular resistance, as well as to the forearm vasodilation that is evoked by bradykinin in vivo [201]. We have also shown that in pre-contracted porcine coronary artery, bradykinin induced deep relaxation was mediated via EDHF, a mechanism that was independent of NO, arachidonic acid metabolism, or reactive oxygen species [202]. ACE inhibition-induced renal vasodilation, which is mediated in part through B_2_ kinin receptor, appears to be dependent on the release of EDHF; this was demonstrated in a canine renal microcirculation in superficial and juxtaglomerular nephrons in an in vivo, in situ, and intact setting [203]. Taken together, these findings suggest that kinins acting on the B_2_ receptors as mediated through endothelium-released factors play an important role in the cardioprotective action of ACE inhibitors. Ac-SDKP is another endogenous peptide that could participate in the protective effects of ACE inhibitors since the circulating concentrations or tissue contents of Ac-SDKP are increased multi-fold in human and rats treated with ACE inhibitor [189,204,205,206]. We and others have demonstrated that part of cardiac and/or renal protective effects could be mediated through Ac-SDKP in experimentally-induced hypertension or diabetes [191,204,207]. Hence, the protective effects of ACE inhibitors are not limited to reduced Ang II production but could be mediated in part through kinins and/or Ac-SDKP by preventing their degradation and increasing their respective circulating and tissue half-life.

Of great interest, the PARADIGM-HF clinical trial showed that angiotensin-neprilysin inhibition was superior to the ACE inhibitor enalapril in patients with heart failure with reduced ejection fraction [225]. The combination drugs lead to reduction in the risks of death and of hospitalization in heart failure patients with reduced ejection fraction. Neprilysin, a neutral endopeptidase, which degrades kinins, enkephalins, natriuretic peptides, and adrenomedullin [53,54], increases the levels of these substances, leading to less vasoconstriction, sodium retention, and maladaptive remodeling. This study clearly illustrates that the combined inhibition of the renin–angiotensin system and neprilysin had effects that were superior to ACEi alone. However, the PARAGON-HF clinical trial showed that angiotensin-neprilysin inhibition failed to deliver the desired decrease in mortality or hospitalization in patients with an ejection fraction that was greater than 45% [226]. Hence, angiotensin-neprilysin inhibition is effective in patients with reduced ejection fraction and not in preserved ejection fraction. These effects could be mediated in part through kinins. Indeed, several studies attempted to demonstrate the dependence of the cardiovascular and renal protective effects of neprilysin inhibitors on kinins by using either bradykinin receptor antagonists, anti-bradykinin antibodies, or serine protease (kallikrein) inhibitors [227]. Two different mechanisms that may account for the potentiation of bradykinin receptor-mediated actions by neprilysin inhibitors have been proposed, including (1) neprilysin inhibitors may potentiate bradykinin receptor-mediated actions by inhibiting bradykinin degradation and increasing bradykinin levels in the vicinity of the receptor, and (2) neprilysin inhibitors may potentiate bradykinin receptor-mediated actions by promoting cross-talk between the neprilysin-inhibitor complex and the bradykinin receptor; this is similar to the cross-talk between the ACE-inhibitor complex and the B_2_ receptor proposed to mediate ACE inhibitor-induced potentiation of bradykinin receptor-mediated effects (see review by Campbell for further details [227]).

## 9. Role of Kinins in the Cardioprotective Effect of ARBs

Blockade of the Ang II type 1 receptor (AT_1_), using ARBs, presents a critical pathway towards achieving antihypertensive and in organ protection. In parallel, activation of the Ang II type 2 receptor (AT_2_) is cardioprotective, through in part the release of kinins and nitric oxide/cylic guanylate monophosphate (NO/cGMP) [228,229,230]. Moreover, we have demonstrated that activated AT_2_ receptors lead to the activation of prolylcarboxypeptidase (PRCP, a plasma pre-kallikrein activator) and release of bradykinin [231]. Both in vitro and in vivo studies have demonstrated that Ang II via the AT_2_ stimulates NO/cGMP production in the vasculature since these effects are blocked by either an AT_2_ or kinin B_2_ antagonist [153,228,232]. Since blockade of AT_1_ increases Ang II, which in turn may activate AT_2_, it seems reasonable that the cardioprotective effect of ARBs is mediated in part by kinins via activation of AT_2_. In fact, studies have shown that ARBs improved cardiac function and ameliorated remodeling in rats with HF post-MI and these effects were attenuated by an AT_2_ or B_2_ antagonist [91] or in mice lacking AT_2_ receptors (AT_2_^-/-^) [233]. Other studies using B_1_^-/-^, B_2_^-/-^ or eNOS^-/-^ mice and kininogen-deficient rats have reported that the lack of kinins or endothelium-derived NO diminished the cardioprotective effect of ARBs [89,95,197,234].

## 10. Material and Methods

### 10.1. For Renovascular Hypertension in B_2_^-/-^ Mice

Male B_2_R-/- (B6; 129S7-bdkrb2*^tm1Jfh/J^*; stock number 002641; 8 weeks old) on a C57BL/6J background were purchased from Jackson Laboratories (Bar Harbor, ME, USA). Mice were housed in an air-conditioned room with a 12-h light/dark cycle and given standard chow and tap water. This study was approved by the Henry Ford Hospital Institutional Animal Care and Use Committee (IACUC). All animal experiments were conducted in accordance with the National Institutes of Health (NIH) Guide for the Care and Use of Laboratory Animals.

### 10.2. Induction of 2K-1C Hypertension

One week after adapting to their new environment, mice were anesthetized with Nembutal (50 mg/kg; i.p.), and the left kidney was exposed through a flank incision. After separating the renal artery and vein, a hand-made silver clip with an internal diameter of 127 µm was placed around the renal artery [235,236]. In the sham operation, the mice had the same surgery, but the artery was not clipped. The experiment was continued for up to 9 weeks. Vehicle or an ACE inhibitor (ACEi) ramipril (1 mg/kg/day) was started in drinking water at week 5 post-surgery (Figure 4).

### 10.3. Systolic Blood Pressure (SBP)

SBP was measured weekly in conscious mice using a noninvasive computerized tail-cuff system (BP-2000, Visitech, Apex, NC, USA). Each SBP reading comprised three sets of 10 measurements, with each set including more than 6 out of 10 successful measurements. Weekly SBP was averaged every 4 weeks.

### 10.4. Data Analysis

All data are expressed as mean ± SE. Student’s two–sample *t*-test was used to compare differences between treatments within the mouse strain.

## 11. Conclusions

We conclude that kinins do not play a fundamental role in the pathogenesis of hypertension, since humans, rats, and mice that are deficient in one or more components of the KKS or chronic KKS blockade do not cause hypertension. Renal kinins help regulate papillary blood flow and water and sodium excretion, which explains why B_2_-KO mice are more salt-sensitive. Kinins are also potent mediators of inflammation by mediating the cardinal signs of inflammation, acting mainly via inducible B_1_ and in certain diseases B_2_. While kinins participate in the acute antihypertensive effect of ACE inhibitors, in general they are not involved in their chronic effects except for mineralocorticoid-salt-induced hypertension. Kinins acting via NO enhance the vascular protective effect of ACE inhibitors during neointima formation. In myocardial infarction produced by ischemia/reperfusion, kinins play an important role in the infarct reduction seen after preconditioning or ACEi treatment. In HF secondary to infarction, the therapeutic effects of ACEi are partially mediated by kinins via NO while that of ARBs is due in part to the activation of AT_2_ via kinins and NO. Thus, kinins play an important role in regulating thermoregulation, cardiovascular and renal function as well as many of the beneficial effects of ACEi and ARBs.

## Figures and Tables

**Figure 1 pharmaceuticals-13-00347-f001:**
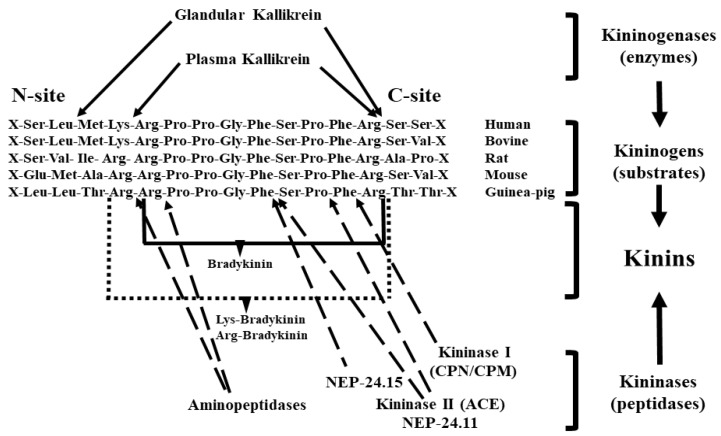
Site of kininogen cleavage (solid arrows) by the main kininogenases (glandular and plasma kallikrein). The broken arrows indicate sites of kinin cleavage by kininases (kininase I, kininase II, neutral endopeptidases 24.11 and 24.15 and aminopeptidases). (Modified after Rhaleb et al. [2,3])

**Figure 2 pharmaceuticals-13-00347-f002:**
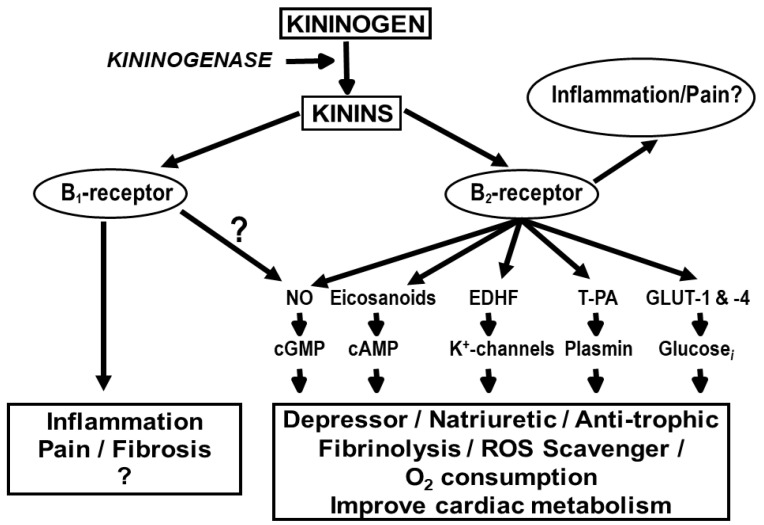
Kinins act via the B_2_ and B_1_ receptors. Most of the known effects of kinins are mediated by the B_2_ receptor which in terms act by stimulating the release of various intermediaries: eicosanoids, endothelium-derived hyperpolarizing factor (EDHF), nitric oxide (NO), tissue plasminogen activator (T-PA), glucose transporter (GLU-1 and -2) (modified from Rhaleb et al. [2])

**Figure 3 pharmaceuticals-13-00347-f003:**
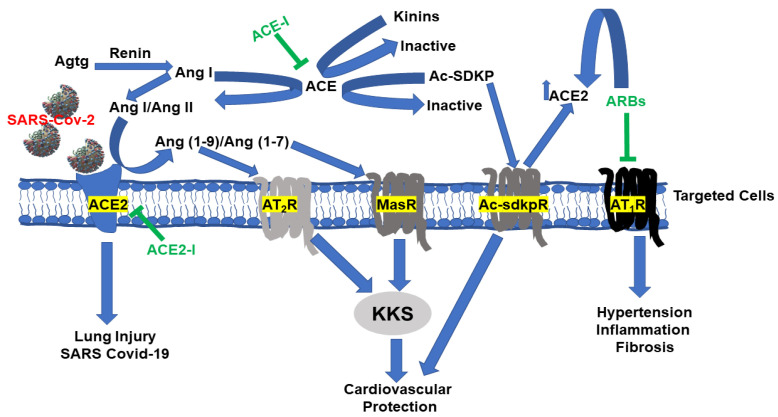
Relationship between RAS and SARS-CoV2: Angiotensinogen (Agtg) is converted to Angiotensin I (Ang I) by renin, which in turn is converted to Ang II by ACE or Ang (1–9) by ACE2 during ACE-I treatment. Ang II is also converted by ACE2 to Ang (1–7) during ARB. ACE-Is and ARBs increase ACE2 expression and activity in animal and human studies through mechanisms that remain to be elucidated. In addition, ACE-I increases circulating and tissue Ac-SDKP, which in turn increases ACE2. Ang II acting through AT_1_ receptor mediate most of the detrimental cardiovascular effects of Ang II through AT_1_ receptors (AT_1_R). Those effects are blocked by ACE-I or angiotensin receptor blockers (ARBs). Activation of AT_2_ by Ang (1–9) or mitochondrial assembly receptor (MasR) by Ang (1–7) mediate some of the protective effects of ACE-I and ARBs). A large population of hypertensive patients is treated with either ACE-I or ARBs, making them at high risk for SARS-CoV-2 associated morbidity and mortality. Binding of ACE2 to SARS-CoV2 leads to viral entry and replication, leading to severe lung injury. ACE2 also degrades desArg^9^-BK but not BK. Potential therapeutic approaches include a SARS-CoV-2 spike protein-based vaccine, blocking the surface ACE2 receptor by using an ACE2 inhibitor, or use of B_1_ receptor antagonists during the period of the propagation of the virus to halt viral spread and the lung or other organs from injury.

**Figure 4 pharmaceuticals-13-00347-f004:**
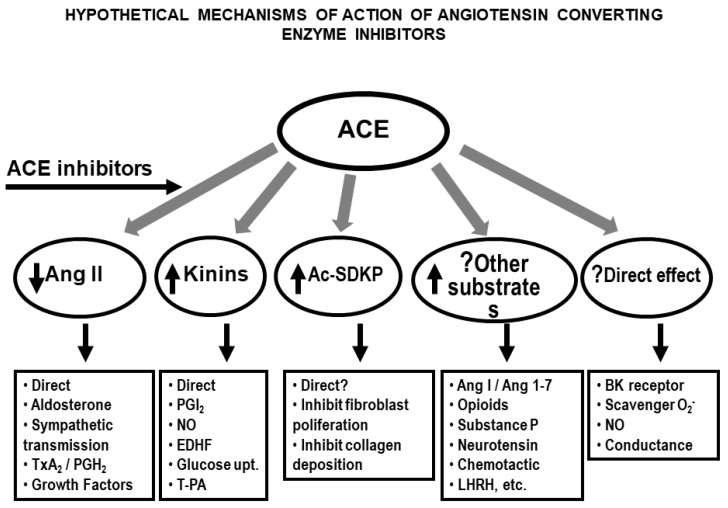
Numerous endogenous peptides are targeted by ACE, resulting in the release of either inactive (for example, kinins and Ac-SDKP) or active ligands such the conversion of Ang I to Ang II. ACE has also been linked to direct effects such as its direct interaction with B_2_ receptors or by scavenging super oxide. Inhibition of ACE resulted in numerous protective effects at the level of the vasculature, heart and kidneys.

**Figure 5 pharmaceuticals-13-00347-f005:**
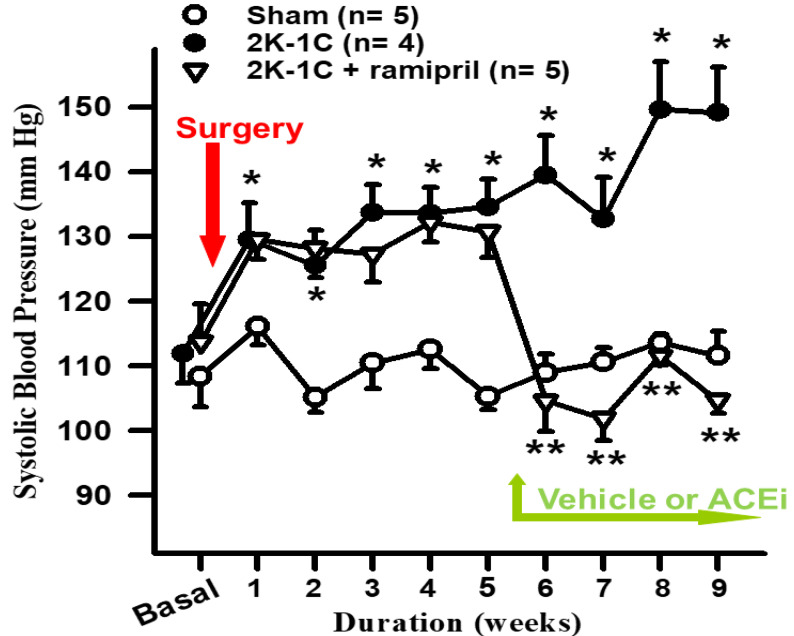
Renovascular hypertension (2 kidney-1 clip) was induced in B_2_^-/-^ mice. At week 5, hypertensive mice were treated with either vehicle or an ACE inhibitor, Ramipril (1 mg/kg/day) in drinking water for 4 weeks. Absence of B2 receptor did not prevent ACE inhibition from normalizing blood pressure in hypertensive mice. *, *p* < 0.05 2K-1C versus Sham; **, *p* < 0.05 2K-1C versus 2K-1C + ramipril. (N.-E., Rhaleb, unpublished observation.)

**Table 1 pharmaceuticals-13-00347-t001:** Role of kinins in myocardial infarction.

Animal Specie	Receptors	Effects	Methods	Tools	References
Mice	B_1_ and B_2_ receptors	Both contribute in cardioprotective effect of ACE-I	MI by left anterior descending coronary artery (LAD) ligation	B_1_-R(-/-) B_2_-R(-/-)	[208]
Mice	B_2_ receptors	Improve cardiac function, tissue remodeling, and inflammation (structural and functional benefits)	MI by LAD ligation	B_2_ receptor selective agonist	[209]
Mice	B_1_ receptors	Cardioprotective effects (improve cardiac function and remodeling)	MI by LAD ligation	B_1_-R(-/-)	[112]
Mice	B_2_ receptors	Cardioprotective effect of ACE-I and ARB	MI by LAD ligation	B_2_-R(-/-)	[95]
Mice	B_2_ receptors	Cardiac remodeling, hypertrophy and dysfunction	MI by LAD ligation	B_2_-R(-/-) B_1_ or AT_2_ receptor antagonist	[88]
Mice	B_1_ receptors	Mediated part of the cardioprotective effects of ACE-I and ARB	MI by LAD ligation	B_1_-R(-/-) B_1_ or AT_2_ receptor antagonist	[112]
Mice	B_2_ receptors	Reduced infarct size reduced cardiomyocyte apoptosis	Ischemia reperfusion	B_1_ and B_2_ receptor agonists, B_2_-R(-/-), B_2_ receptor antagonist, preconditioning	[94,210,211]
Mice	B_2_ receptors	Reduced infarct size and cardio-protection	Ischemia reperfusion	Tissue-kallikrein deficient mice, AT_1_ and AT_2_ receptor antagonists	[212]
Rats	B_2_ receptors	Reduced infarct size	Isolated heart, ischemia reperfusion	Brown Norway Katholiek (BN-Ka) rats, B_1_ or B_2_ receptor antagonists or agonists, neutral endopeptidase (NEP) inhibitor, ACE-I	[94,213,214]
Rats	B_2_ receptors	Inhibits collagen deposition, reduce myocardial collagen accumulation by ACE-I and ARB	MI by LAD ligation	B_2_ receptor antagonist	[215]
Rats	B_2_ receptors	Mediated protective effects of ARB and ACE-I	MI by LAD ligation	ACE-I, ARBs, AT_2_ receptor antagonist	[91]
Rats	B_2_ receptors	Reduce infarct size	MI by LAD ligation	BN-Ka rats, B_2_ receptor antagonist, and a nonpeptide B_2_ receptor agonist	[216]
BN-Ka	kininogen	Kinin do not mediate the beneficial effects of ACE-I	MI by LAD ligation	BN-Ka versus BN Norway Hannover (wild-type rats)	[217]
Rat	B_2_ receptors	Inhibit the interstitial accumulation of collagen, no effects on cardio myocyte hypertrophy	Morphometric analysis, collagen deposition in left ventricular interstitial	B_2_ receptor antagonist	[218]
Rats	B_1_ receptors	Inhibited myocardial noradrenaline, reduced ventricular fibrillation	Ischemia reperfusion	B_1_ receptor agonist and antagonist	[219]
Rabbit	B_2_ receptors	Reduction in infarction size	Ischemia/reperfusion	B_2_ receptor antagonist	[220]
Rabbit	B_2_ receptors	Mediated the effect of ACE-I on infarct size	MI by LAD ligation plus high cholesterol diet	B_2_ receptor antagonist	[221]
Dog	B_1_-receptor	Hypotensive effect, peripheral vasodilation	Intra-arterial and intravenous injection	B_1_ agonist	[222]
Dogs	B_1_ and B_2_ receptors	Decreases mean arterial pressure (MAP) and coronary vascular resistance (CVR)	i.v. infusion	B_1_ receptor agonist and antagonist	[223]
Dogs	B_2_ receptors	Reduced infarct size	Isolated heart, ischemia reperfusion	Combined NEP/ACE inhibitor, B_2_ receptor antagonist	[224]

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
