# Peer review of "Role of Kinins in Hypertension and Heart Failure"

_pharmaceuticals, 2020, doi:10.3390/ph13110347_

Round 1
Reviewer 1 Report
In this manuscript, Hamid et al. provide a review of the available data regarding the role of kinins in hypertension and heart failure. This is a valuable review of the above topic. This review is well conceived and structured, and provides current knowledge in this area (including authors' own work). This reviewer has only a few minor comments listed below.
One aspect that this reviewer would suggest to address is possible effects of bradykinin on sympathetic nerves innervating blood vessels. Kansui et al, for example, showed that bradykinin enhanced sympathetic neurotransmission and that an ACE inhibitor captopril enhanced the action of bradykinin on sympathetic neurotransmission in rat mesenteric arteries. Such an effect might partially counteract bradykinin induced vasodilation.
It would also be advisable to elaborate more on the possible involvement of EDHF in bradykinin induced vasodilation. Since EDHF appears to serve as a backup vasodilator when NO-mediated vasorelaxation is reduced, bradykinin-induced, EDHF-mediated vasorelaxation could play a crucial role in the maintenance of endothelial function in disease states such as hypertension and heart failure where NO-mediated responses are compromised due to increased oxidative stress.
Trivial
Abstract lines 18-22. Unify the font size
Page 5, line 155. “ae” should read “are”
Page 5, line 176. “ANF” should read “ANP”
Page 6, line 214. “the role” is duplicated.
Page 11, line 393. Delete “McMurray”
Page 11, line 401. Delete “Solomon”
Author Response
First of all, thank the reviewer's compliment and comments:
We have accepted all suggestions from this reviewer, including the insertion of a Table in which we have depicted the possible role of kinins in myocardial infarction. We al made the minor corrections. All changes are made in red font.
Reviewer 2 Report
In this review, the authors summarized current knowledge about the important role of kinins in hypertension, focusing on heart failure. I think that the present manuscript has a complete, interesting and current approach and could be a starting point for a broader analysis for a better understanding of the possible antihypertensive implications of kinins. My overall impression is that this good review would be of great interest to researchers working in the hypertension field and beyond.
I have only one comment, it is necessary a table that summarizes the role of kinins in myocardial infarction.
Author Response
Thank you for the reviewer's comment and per her(his) suggestion we have inserted a Table in which we have summarized possible role of kinins in myocardial infarction.